# Outcomes in Pregnant Persons Immunized with a Cell-Based Quadrivalent Inactivated Influenza Vaccine: A Prospective Observational Cohort Study

**DOI:** 10.3390/vaccines10101600

**Published:** 2022-09-23

**Authors:** Christopher Robinson, Josephine Van Boxmeer, Hugh Tilson, Anthony Scialli, John A. Vanchiere, Ellis Ides, Daphne Sawlwin, Deborah Molrine, Matthew Hohenboken, Jonathan Edelman, Jessica D. Albano

**Affiliations:** 1Charleston Maternal Fetal Medicine, Mount Pleasant, SC 29485, USA; 2Seqirus Netherlands B.V., 1105 BJ Amsterdam, The Netherlands; 3Gillings School of Global Public Health, University of North Carolina, Chapel Hill, NC 27599, USA; 4Scialli Consulting LLC, Washington, DC 20008, USA; 5Louisiana State University Health Science Center, Shreveport, LA 71106, USA; 6Seqirus Pty Ltd., Parkville, VIC 3052, Australia; 7Seqirus USA Inc., Cambridge, MA 02139, USA; 8Seqirus USA Inc., Summit, NJ 07901, USA; 9Syneos Health, Wilmington, NC 28403, USA

**Keywords:** cell-based quadrivalent influenza vaccine, stillbirth, spontaneous abortion, preterm birth, low birth weight, major congenital malformations, vaccination

## Abstract

**Objective:** To evaluate pregnancy and infant outcomes among persons immunized with a cell-based quadrivalent inactivated influenza vaccine (IIV4c) during routine pregnancy care. **Design****:** Prospective observational cohort. **Setting****:** US-based obstetrics/gynecology clinics. **Population:** Pregnant persons. This US-based, prospective observational cohort study evaluated the safety of quadrivalent inactivated influenza vaccine (IIV4c; Flucelvax^®^ Quad) in pregnant persons immunized over 3 influenza seasons between 2017 and 2020. Pregnant persons were immunized with IIV4c as part of routine care, after which their health care provides HCPs with all observational data to a single coordinating center. Follow-up data were collected at the end of the second trimester and/or at the time of pregnancy outcome. A scientific advisory committee reviewed the data. Prevalence point estimates were reported with 95% confidence intervals (CIs). Pregnancy outcomes included: live birth, stillbirth, spontaneous abortion, elective termination, and maternal death. Infant outcomes included: preterm birth (<37 weeks gestational age), low birth weight (<2500 g), or major congenital malformations (MCMs). Of the 665 evaluable participants, 659 (99.1%) had a live birth. No stillbirths (0% [95% CI 0.0–0.6]), 4 spontaneous abortions (1.9% [0.5–4.8]), and 1 elective termination (0.5% [0.0–2.6]) were reported. Among 673 infants, 9.2% (upper 95% CI 11.5%) were born prematurely, 5.8% (upper 95% CI 7.6%) had low birth weight, and 1.9% (upper 95% CI 3.1%) were reported to have an MCM. No maternal deaths were reported. Of the 2 infants who died shortly after birth, one was adjudicated as not related to the vaccine; the other’s cause could not be determined due to maternal loss to follow-up. The prevalence of adverse pregnancy outcomes or preterm birth, low birth weight, or MCMs in newborns was similar in persons vaccinated with IIV4c compared to the rates observed in US surveillance systems. The safety profile of IIV4c in pregnant persons is consistent with previously studied influenza vaccines.

## 1. Introduction

Pregnant persons are considered a priority group for seasonal influenza vaccination to prevent influenza-related complications, among other at-risk groups such as older adults, very young children and persons with certain chronic medical conditions [1,2,3]. Pregnant persons are at increased risk for morbidity and death during seasonal influenza epidemics and influenza pandemics, and newborn infants born to persons who had influenza during pregnancy—especially with severe illness—are at increased risk of adverse outcomes, including preterm birth and low birth weight [4,5]. Moreover, infants < 6 months of age who experience influenza virus infection have the highest rates of hospitalization and death of all children [6]. The safety of influenza vaccinations during pregnancy is well documented [7,8,9,10,11], yet only 61% of pregnant persons received an influenza vaccination in 2019 [12]. Among other reasons, safety concerns for the fetus are frequently cited as barriers to vaccination among pregnant persons [11].

The majority of inactivated influenza vaccines, including those evaluated in previous studies with pregnant persons, are produced in embryonated chicken eggs. The viruses within these vaccines are susceptible to selection-induced, adaptive mutations in the viral hemagglutinin protein, which can alter the antigenicity and effectiveness of the vaccines [13,14,15,16]. In contrast, by avoiding growth in eggs, vaccine viruses propagated in mammalian cell cultures can yield vaccine strains that are a closer antigenic match to the original virus [17,18,19]. The avoidance of egg-adaptive mutations may lead to increased effectiveness of cell culture-derived vaccines compared with egg-derived vaccines, as suggested by observational studies [20,21,22,23,24,25]. A cell culture-based manufacturing platform offers an improved production process and a more rapid surge capacity in the event of a pandemic. A cell-based inactivated quadrivalent influenza vaccine prepared from viruses propagated in Madin-Darby Canine Kidney (MDCK) cells (IIV4c; Flucelvax Quadrivalent, Seqirus USA Inc., Holly Springs, NC, USA) was approved in the US in May 2016 for use in persons 4 years of age and older [26]. Data on safety in pregnant persons, however, are limited [1]. To address this gap, we evaluated safety outcomes in pregnant persons immunized with IIV4c during routine obstetric care by prospectively collecting data on pregnancy and infant outcomes, including major congenital malformations, preterm birth, and low birth weight.

## 2. Materials and Methods

### 2.1. Study Design

This US-based, prospective observational cohort study was designed as a pregnancy exposure registry in line with the US Food and Drug Administration (FDA) guidance [27] and conducted to fulfill a post-marketing commitment to FDA. The study was strictly observational; the schedule of visits, the laboratory tests, all treatment regimens, and the timing of immunizations were determined by the treating health care provider (HCP). The Syneos Health Registry Coordinating Center (Wilmington, NC, USA) coordinated enrollment and data collection and served as the single site for the study. Study data originated from persons who self-enrolled or were enrolled from obstetrics/gynecology clinics that used IIV4c as part of routine care to vaccinate pregnant persons.

This study was conducted in accordance with the Good Pharmacoepidemiology Practices, applicable local regulations, and the Declaration of Helsinki [28,29]. An Institutional Review Board approved the protocol and consent form. All participants provided either written or verbal informed consent. Because this study involved routine care that posed no excess risk to participants, written consent was not required per US regulations. Protected health information and identifying information for all study participants were removed prior to data reporting.

### 2.2. Study Population

The study population included pregnant persons residing within the US who were immunized with IIV4c as part of routine care at any time during pregnancy. Participants were enrolled prospectively by obstetric HCPs who had committed to enrolling all eligible persons in their clinic who consented to the study; pregnant persons could also self-enroll during their pregnancies, after which their HCPs provided all pertinent data to the registry (see the Appendix A for full information on participant selection and recruitment). The evaluable population included participants for whom sufficient data were available to confirm the date of receipt of the vaccine, to determine the pregnancy outcome and to evaluate whether an event of interest had occurred for the fetus/infant at the time of the pregnancy outcome. Retrospective cases were ineligible for enrollment, as were persons who had prior knowledge of an adverse pregnancy outcome (such as an MCM suggested by prenatal testing). These cases were reported to the sponsor’s routine pharmacovigilance system.

### 2.3. Vaccination

This study was conducted with Flucelvax Quadrivalent, which was formulated to contain hemagglutinin antigens from the following 4 influenza strains, selected for the season of vaccination: A/H1N1, A/H3N2, B/Yamagata, and B/Victoria. Vaccination was not part of the protocol and study procedures; instead, the vaccine was administered solely as part of routine clinical care at the discretion of the pregnant person and the treating HCP.

### 2.4. Pregnancy Outcomes and Events of Interest

Pregnancy outcomes were reported as live birth, stillbirth (fetal death occurring at ≥20 weeks of gestation or a fetus weighing ≥ 500 g), spontaneous abortion (fetal death prior to 20 weeks of gestation), or elective termination. Ectopic pregnancy or molar pregnancy were reported as pregnancy types. Specific events of interest for fetuses/infants included preterm birth (infants born at <37 weeks gestation), low birth weight (<2500 g), or major congenital malformations (MCMs). Outcomes of interest included any major structural or chromosomal defect or a combination of three or more minor birth defects in live-born infants, stillbirths, and fetal losses at any gestational age, which closely resembles the definition used by the Centers for Disease Control and Prevention (CDC) Metropolitan Atlanta Congenital Defects Program (MACDP) [30]. Follow-up ended upon completion of pregnancy.

No separate assessments were required for the study. Outcome data, including gestational age and birth weight, were documented in the participants’ medical records as part of routine care and were provided to the registry coordinating center by participating HCPs (including obstetricians or, when needed, pediatricians attending the birth) or pregnant persons postpartum. A teratologist who was blinded to the timing of vaccination reviewed all reported congenital malformations, classified them using the MACDP coding system, and provided an opinion regarding the gestational time window during which exposure to IIV4c could have potentially had an impact on the reported birth defect. A scientific advisory committee (SAC) comprised of experts in obstetrics/maternal-fetal medicine, pediatrics, clinical research, infectious disease, epidemiology, and teratology met at least annually to review and classify reported MCMs and review aggregated data on other events of interest and provide an interpretation of the data.

### 2.5. Statistical Methods

A sample size of 660 enrolled individuals was planned to allow for approximately 10% of the study population to be lost to follow-up, leaving at least 600 evaluable participants in the study population. This sample size would provide ≥ 80% power to detect a 2.4-fold increase in MCMs compared with the background prevalence of 2.78% [31], assuming that one-third of the study population (*n* = 200) would be exposed to the vaccine during the first trimester. It would also provide >99% power to detect 2.0-fold increases in preterm birth and low birth weight, which had background prevalences of 11.39% and 8.02%, respectively, at the time the study was designed [32].

Only descriptive statistics were used in this study. Overall and stratum-specific prevalence point estimates and 1-sided (upper) 95% confidence intervals (CIs) were calculated using the exact binomial distribution for the following: the prevalence of MCMs among all evaluable live-born infants (in line with the MACDP analysis method); the prevalence of low birth weight among evaluable persons with a singleton live birth and an infant without an MCM; and the prevalence of preterm birth among evaluable participants enrolled prior to 37 weeks of gestation with a singleton live birth and an infant without an MCM. The prevalence of observed MCMs, preterm birth, and low birth weight were compared to recently reported prevalence estimates from the CDC MACDP, National Center for Health Statistics (NCHS), and National Vital Statistics System (NVSS), respectively [31,33,34]. Analyses were stratified by trimester of exposure.

## 3. Results

Of the 693 participants enrolled, 1 person was deemed ineligible after enrollment, and 27 (3.9%) were lost to follow-up. The primary analysis population consisted of 665 evaluable participants who received IIV4c as part of their routine pregnancy care during one of three Northern Hemisphere influenza seasons (2017–2018, 2018–2019, and 2019–2020). Except for 9 participants who self-enrolled, all participants were enrolled by 5 obstetric HCPs, located in Idaho, New York, Georgia and North Carolina. Of the primary analysis population, 31.7% were enrolled prior to 20 weeks of gestation; 26.8% received IIV4c during the first trimester, 41.7% received IIV4c during the second trimester, and 31.6% received IIV4c during the third trimester (Appendix A). The earliest exposure to IIV4c was at 5 weeks of gestation, and the latest was at 39 weeks (Appendix A).

There were no major differences in the demographic and baseline characteristics between the persons enrolled by the different HCPs. The mean maternal age at conception was 28 years, and most participants were between 25 and 34 years of age. Although most participants were non-Hispanic whites, the population was diverse and generally representative of the overall US population. Most participants had at least one concurrent medical condition with onset before or during pregnancy and were taking medication, including prenatal vitamins (Appendix A). Tobacco use was reported by 84 (12.6%) participants, and 1 reported alcohol use. No participants reported using illicit drugs. Eight participants (1.2%) reported having a previous child with a congenital malformation, and 11.7% had a family history of congenital malformations. The demographic characteristics of the participants lost to follow-up were similar to those of the population that completed the study.

### Outcomes

Table 1 lists the pregnancy outcomes by trimester of exposure. A total of 659 out of 665 (99.1%) evaluable participants (Primary Analysis Population) had a live birth. No stillbirths were reported. Among the 211 persons enrolled during the first 20 weeks of gestation, 1 elective termination was reported, and there were 4 spontaneous abortions, 1 each in the 20–24-year and ≥40-year age groups and 2 in the 25–34-year age group. One ectopic and no molar pregnancy were reported.

Data were reported for 673 infants, including 656 singletons and 9 pairs of twins (Table 2). For one set of twins, one of the twins died in utero prior to vaccination and enrollment and was therefore not included in the infant dataset. In the population used to calculate the prevalence of preterm birth (see Statistical Methods), 52 infants were born preterm (9.2%, upper 95% CI 11.5%), and among the population used to calculate the low birth weight prevalence, a birth weight of <2500 g was reported for 37 infants (5.8%, upper 95% CI 7.6%) (Figure 1).

For 17 fetuses and infants at least one MCM was reported by an HCP, and, 14 were adjudicated by the scientific advisory committee as having a confirmed MACDP defect, and at least one MCM was reported by an HCP. Out of this group, 1 birth defect was observed in a pregnancy that was electively terminated prior to 20 weeks of gestation. The prevalence of MCMs was 13 out of 667 live-born infants, or 1.9% (upper 95% CI 3.1%) (Figure 1). Vaccine exposure during the first trimester was reported for 2 of the 14 infants with birth defects adjudicated as MACDP defects (including the defect identified in the electively terminated pregnancy; Table 3). In both cases, the reported birth defect had a known cause other than exposure to the study vaccine.

Two deaths were reported for live-born infants. One infant who died 24 h after birth had polycystic kidneys, and fetal anhydramnios was reported during pregnancy. Fetal polycystic kidneys were assessed as an MACDP MCM but did not have a temporal association with the vaccination, which took place at 16.1 weeks of gestation. The other infant died 2 days after birth. No information on the cause of death or MCMs was available.

## 4. Discussion

### 4.1. Main Findings

In this prospective observational cohort study of pregnant persons vaccinated with IIV4c, the frequency of adverse pregnancy outcomes was not increased when compared to the rate in the general US population. Of 665 pregnancies, 99.1% ended with a live birth. There were 4 spontaneous abortions, 1 elective termination due to an MCM, 1 ectopic pregnancy and no stillbirths. The prevalence of preterm birth, low birth weight, and MCMs was similar to the prevalence reported in the literature for the general US population [31,33,35].

### 4.2. Strengths and Limitations

Data were collected across multiple influenza seasons and from various geographical locations throughout the US, although the majority of participants were from 2 states, North Carolina and Georgia. The study population included the racial and ethnic groups most prevalent in the US general population, as well as a broad range of maternal ages and numbers of previous pregnancies. Almost 30% of the persons enrolled had a first trimester exposure, which is important for the assessment of maternal exposures and birth defects. Both high-risk and low-risk pregnancies were included in the study, although there is a chance that very high-risk pregnancies may not have been under the prenatal care of the OB/GYN clinics that participated.

To avoid selection bias toward more adverse outcomes, the study excluded persons who had prior knowledge of an adverse pregnancy outcome (such as an MCM suggested by prenatal testing) at the time of enrollment. This practice, however, could have potentially biased the results toward lowering the overall risk of MCMs in the study population. The ability to examine the effect of potential confounders and effect modifiers, such as previous pregnancy outcomes, pregnancy complications, concurrent conditions, and concomitant exposures, was also limited because the enrolled numbers were too small for meaningful stratified analyses and because of the large variation in the type of information reported for each of the prespecified categories. Therefore, we could not implement a multivariate analysis with the aim to investigate the potential confounding effects of these variables.

Twenty-seven persons (3.9%) were lost to follow-up, which limited the bias induced by this category of participant disposition. For 5 persons, although data on pregnancy outcomes were provided (all 5 were live births), the presence or absence of an MCM could not be confirmed. For the remaining 22 persons, 17 of whom left the care of their HCP after having completed the first trimester of pregnancy, the reason for loss to follow-up was no longer being under the care of a participating HCP. Although it could not be confirmed why each pregnant person left the participating HCP clinic, 6 persons moved out of state, and 2 were being referred to more specialized care. Although it cannot be established that the pregnancy outcome data for this small population followed the same patterns seen in the evaluable study population, the percentage lost to follow-up is within the range reported from a recent review of prospective pregnancy registries [36].

When observational data are collected by an HCP as part of routine care, there is a risk of missing data or a limited level of detail, lower precision of measures, and potential ascertainment bias of adverse pregnancy outcomes (e.g., the reporting HCP may not know the condition of an aborted fetus). Finally, given the lack of a concurrent comparator group, we cannot rule out the effect of potential differences between the exposed study population and the broader US population used for comparative statistics. Our study used the same definitions for the events of interest as the sources of data used for comparison, including the CDC MACDP [30]. However, the MACDP is an active surveillance program and therefore includes all birth defects reported in the defined region, whereas participation in the study was on a voluntary basis. Furthermore, MACDP counts any MCMs that are reported in an infant up to the age of 6 years, while our study conducted follow-up only until pregnancy was completed. There is a chance that major MCMs that are not immediately detectable at birth might have been missed due to this chosen methodology.

### 4.3. Interpretation

The 95% CI for stillbirths in our study was 0.0–0.6, the upper limit of which is consistent with the overall US fetal mortality rate of 0.6% [34]. We observed a rate of spontaneous abortion of 1.9% (95% CI 0.5–4.8), which is considerably lower than the estimated rate of 10% from the American College of Obstetrics and Gynecologists (ACOG) [37]. Although only persons enrolled prior to 20 weeks of gestation were included in the denominator for the spontaneous abortion calculation, the risk of spontaneous abortion decreases with increasing gestational age and is the highest early in the first trimester. No persons were enrolled in the study prior to 5 weeks of gestation, which may contribute to the lower prevalence of spontaneous abortion in our study.

The preterm birth prevalence observed in this study (9.2% [upper 95% CI 11.5%]) was consistent with the preterm birth rate of 10.23% in the general US population in 2019 (Figure 1) [33]. Our study enrolled a relatively large sample of black persons (comprising 30% of the study population, compared with 13% of the general US population) [38]. This difference may have affected the prevalence of preterm birth in our study because the preterm birth rate is approximately 50% higher in non-Hispanic Black persons than in non-Hispanic White persons in the US [33].

The 2018 US prevalence of low birth weight for all births was 8.28% with a rate of 6.60% reported for singleton births [35]. The point estimate (5.8%) and the 95% upper CI (7.6%) observed in our study are consistent with both rates of the general US population, with the rate for singleton births a more direct comparison based on the study calculation used for this infant outcome.

The point estimate for the prevalence of MCMs in our study (1.9% [upper 95% CI 3.1%]) was similar to the background prevalence of 2.78% [31]. Although the upper 95% CI is higher than the population statistic, the CI is wide due to the sample size of the study. Upon review of the individual reports as well as aggregated data, no patterns were found among the reported events.

Previous studies have not found evidence that maternal immunization increases the risk of adverse pregnancy outcomes [2,7,8,9,10,11], and seasonal influenza vaccination is recommended to prevent influenza-related morbidity and mortality in both pregnant persons and their infants [1,2]. Between 2010 and 2019, 28% of persons of childbearing potential who were hospitalized for influenza were pregnant, of which 5% were admitted to intensive care units, 2% required mechanical ventilation, and 0.3% died. In addition, 3% of the patients who were no longer pregnant at discharge had experienced fetal loss [5]. Despite the risk of the severe health impacts of influenza, many pregnant persons cite safety concerns as a reason for vaccine refusal, and the 61% vaccination rate for pregnant persons in 2019 fell short of the Healthy People 2020 recommendation of 80% [11,12,39]. Because cell-based influenza vaccines are produced via a different manufacturing process than egg-based vaccines, efforts to ascertain the safety of newer formulations are necessary to reassure pregnant persons and their health care providers of the absence of new or increased risks. Our data were collected from a study population that included persons of diverse racial and ethnic ancestry as well as an age range that spanned the years of childbearing potential. Importantly, vaccinations also occurred across gestational weeks of pregnancy. Our findings suggest that there is no safety concern of exposure to IIV4c during pregnancy.

Our data add to the overall body of evidence that inactivated influenza vaccines are safe for use in pregnant persons. Ongoing routine monitoring will provide further continuous reassurance.

## 5. Conclusions

Our findings are consistent with published data from various databases and surveillance systems that monitor the safety of influenza vaccines in pregnant persons. These data support the use of IIV4c for immunization against influenza in this population.

## Figures and Tables

**Figure 1 vaccines-10-01600-f001:**
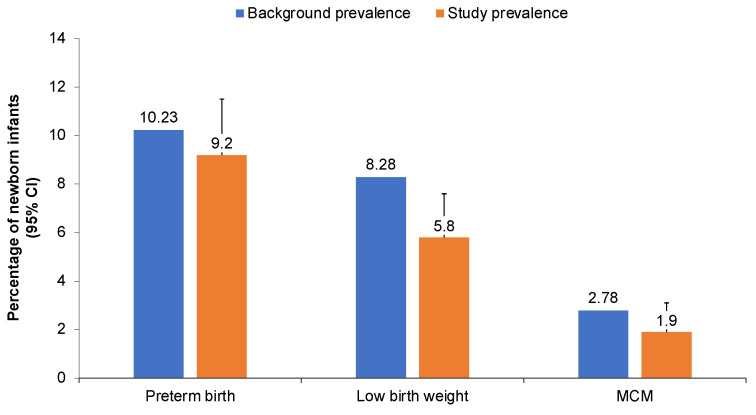
Prevalence of adverse infant outcomes in the general (background) and study populations for preterm birth, low birth weight, and major congenital malformations (MCMs) [33,35]. Error bars represent the upper limit of the 95% confidence interval (CI).

**Table 1 vaccines-10-01600-t001:** Pregnancy outcomes.

Outcome, *n*, % (95% CI)	Vaccine Exposure			Overall
	First Trimester ^a^	Second Trimester	Third Trimester	
All enrolled persons	*n* = 196	*n* = 286	*n* = 211	N = 693
Primary analysis population	*n* = 178	*n* = 277	*n* = 210	N = 665
Enrollment < 20 weeks of gestation	*n* = 147	*n* = 64	N/A	N = 211
Live birth, *n* % (95% CI)	172 96.6 (92.8–98.8)	277 100 (98.7–100)	210 100 (98.3–100)	659 99.1 (98.0–99.7)
Stillbirth	0 0 (0.0–2.1)	0 0 (0.0–1.3)	0 0 (0.0–1.7)	0 0 (0.0–0.6)
Spontaneous abortion ^b^	4 2.7 (0.7–6.8)	0 0 (0.0–5.6)	N/A	4 1.9 (0.5–4.8)
Elective termination ^b^	1 0.7 (0.0–3.7)	0 0 (0.0–5.6)	N/A	1 0.5 (0.0–2.6)

^a^ One ectopic pregnancy was reported with a first trimester exposure but was not reported as one of the pregnancy outcomes listed here. ^b^ Calculated using the population enrolled at <20 weeks of gestation (*n* = 211) as the denominator.

**Table 2 vaccines-10-01600-t002:** Infant outcomes.

Outcome	Vaccine Exposure			Overall
	First Trimester ^a^	Second Trimester	Third Trimester	
Persons	*n* = 178	*n* = 277	*n* = 210	*n* = 665
Infants	*n* = 178	*n* = 282	*n* = 212	*n* = 673
Sex, *n* (%)				
Male	84 (47.2)	136 (48.2)	111 (52.4)	331 (49.2)
Female	88 (49.4)	146 (51.8)	101 (47.6)	335 (49.8)
Missing data	6 (3.4)	0	0	7 (1.0) ^a^
Birthweight	*n* = 172	*n* = 278	*n* = 211	*n* = 661
Mean ± SD, g	3295.4 ± 562.3	3236.1 ± 561.3	3313.4 ± 506.3	3276.2 ± 544.9
Mean gestational age at outcome ± SD, weeks	37.9 (5.1)	38.5 (2.1)	39.1 (1.2)	38.5 (3.0)
Preterm birth ^b^	*n* = 166	*n* = 250	*n* = 149	*n* = 565
*n*, % (95% upper CI)	17 10.3 (15.1)	27 10.8 (14.6)	8 5.4 (9.5)	52 9.2 (11.5)
Low birthweight ^c^	*n* = 170	*n* = 264	*n* = 203	*n* = 637
*n*, % (95% upper CI)	14 8.3 (12.6)	15 5.7 (8.6)	8 3.9 (7.0)	37 5.8 (7.6)
MCMs (live-born infants)	*n* = 173	*n* = 282	*n* = 212	*n* = 667
*n*, % (95% upper CI)	1 0.6 (2.7)	7 2.5 (4.6)	5 2.4 (4.9)	13 1.9 (3.1)

Abbreviations: CI, confidence interval, MCM, major congenital malformation; SD, standard deviation. ^a^ Strata do not add up because one participant was missing data for date of outcome and gestational age at outcome. ^b^ Excludes non-live births, live births of infants with MCMs, multiple gestation pregnancies, and any participant enrolled after 37 weeks of gestation. ^c^ Excludes non-live births, live births of infants with MCMs, and multiple gestation pregnancies.

**Table 3 vaccines-10-01600-t003:** MACDP-coded major congenital malformations.

Participant	Timing of Exposure (Week of Gestation)	Preferred Term
1	5.4	Sex chromosome: XYY in a male
2	10.7	Talipes equinovarus
3	16.1	Polycystic kidneys
4	16.1	Renal agenesis, right: MACDP term “Kidney—absence, agenesis, dysplasia, or hypoplasia unilateral, right”
5	16.4	Clubfoot
		Cardiomegaly
		Aorta malformation ^a^
		Hypoplasia of upper limb, hypoplasia of lower limb
6	18.1	Situs inversus abdominus
7	19.9	Hirschsprung’s disease
8	23	Fluid around kidneys ^a^
9	24.9	Micropenis
		Microphthalmos
10	30.3	Transposition of great vessels
11	32.1	Trisomy 21
		Atrial septal defect
		Patent ductus arteriosus
12	33	Absent foreskin
13	33.3	Hypospadias
14	33.4	Absent forearm only, left

Abbreviations: MACDP, Metropolitan Atlanta Congenital Defects Program. ^a^ Insufficient information available to code the reported event as an MACDP Preferred Term. Therefore, the verbatim reported term is provided.

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
