# Peer review of "Outcomes in Pregnant Persons Immunized with a Cell-Based Quadrivalent Inactivated Influenza Vaccine: A Prospective Observational Cohort Study"

_vaccines, 2022, doi:10.3390/vaccines10101600_

Round 1
Reviewer 1 Report
Title
“Outcomes in Pregnant Persons Immunized with a Cell-Based Quadrivalent Inactivated Influenza Vaccine: A Prospective Observational Cohort.
Review
The manuscript studies the outcomes in pregnant mothers immunized with a cell-based 2 quadrivalent inactivated influenza vaccine. The issue of this research is important. The objective of this review is to improve the manuscript, and some suggestions may be addressed.
1. For a better understanding of the study, the participation rate could be mentioned. What proportion of pregnant women participated in the study considering the total of assisted pregnant women in the clinics where the study took place during the study period?
2. The authors mention the prevalence point of the outcomes. However, the study is a prospective cohort study and the incidence of new outcomes is reported on this design (1-2). Could explain the authors this situation?
3. Could explain the use of 1-side (upper) 95% confidence intervals (CIs) instead of 2-side 95% CI?
4. The places of the pregnant women were studied could be indicated for a better comparison with the general US population.
5. Some characteristics of the study population are indicated in the supplements1. However, some variables are missing such as socio-economic level, geographic distributions, and maternal medical condition. In addition, no statistical comparison between participants and the lost population is performed.
6. The authors use the expression “not statistically significantly different” (Page 8, line 306) when comparing outcomes of the general US population. What statistical test was employed?
7. Limitations of the study. The sample size was small to detect significant differences for many of the outcomes, and the absence of a control group is another important limitation as the authors reported. In addition, some outcomes are missing like the Apgar index or small for gestational age. Considering these circumstances, the generalization of the results may be problematic.
References:
1. Eaton A, Lewis N, Fireman B, Hansen J, Baxter R, Gee J, Klein NP. Birth outcomes following immunization of pregnant women with pandemic H1N1 influenza vaccine 2009-2010. Vaccine. 2018 May 3;36(19):2733-2739.
2. Chambers CD, Johnson D, Xu R, Luo Y, Louik C, Mitchell AA, Schatz M, Jones KL; OTIS Collaborative Research Group. Risks and safety of pandemic H1N1 influenza vaccine in pregnancy: birth defects, spontaneous abortion, preterm delivery, and small for gestational age infants. Vaccine. 2013 Oct 17;31(44):5026-32.
Author Response
REVIEWER #1 |
AUTHOR RESPONSE |
The manuscript studies the outcomes in pregnant mothers immunized with a cell-based 2 quadrivalent inactivated influenza vaccine. The issue of this research is important. The objective of this review is to improve the manuscript, and some suggestions may be addressed. |
Thank you for your assessment |
1. For a better understanding of the study, the participation rate could be mentioned. What proportion of pregnant women participated in the study considering the total of assisted pregnant women in the clinics where the study took place during the study period? |
The vast majority of participants in the study were enrolled by so-called ‘dedicated reporters’, which were obstetric HCP who had signed an intention to enroll all eligible persons who consented to the study (see lines 95-96 of the manuscript). Persons who declined consent are not entered into the study, therefore we have no insights into numbers and we cannot provide a participation rate. |
2. The authors mention the prevalence point of the outcomes. However, the study is a prospective cohort study and the incidence of new outcomes is reported on this design (1-2). Could explain the authors this situation?
References: 1. Eaton A, Lewis N, Fireman B, Hansen J, Baxter R, Gee J, Klein NP. Birth outcomes following immunization of pregnant women with pandemic H1N1 influenza vaccine 2009-2010. Vaccine. 2018 May 3;36(19):2733-2739. 2. Chambers CD, Johnson D, Xu R, Luo Y, Louik C, Mitchell AA, Schatz M, Jones KL; OTIS Collaborative Research Group. Risks and safety of pandemic H1N1 influenza vaccine in pregnancy: birth defects, spontaneous abortion, preterm delivery, and small for gestational age infants. Vaccine. 2013 Oct 17;31(44):5026-32. |
Prevalence is used to measure how often certain outcomes occur in a population in a certain time period whereas incidence is used is to measure the number of newly developed cases in a population in a certain time period. It is indeed true that we measured pregnancy and infant outcomes that developed during the study period, however, since predefined outcomes for this study could by definition only occur once, the prevalence and incidence measures are the same. |
3. Could explain the use of 1-side (upper) 95% confidence intervals (CIs) instead of 2-side 95% CI? |
The 1-sided upper CI was chosen since the study is designed to identify any potential safety signal. A safety signal would possibly occur if the prevalence found in the study would be higher than the background population data. |
4. The places of the pregnant women were studied could be indicated for a better comparison with the general US population. |
Thank you. The US states where the HCPs are located are added in line 165 (Idaho, New York, Georgia and North Carolina). |
5. Some characteristics of the study population are indicated in the supplements1. However, some variables are missing such as socio-economic level, geographic distributions, and maternal medical condition. In addition, no statistical comparison between participants and the lost population is performed.
|
For this study, socio-economic level was not collected as part of the demographic and baseline information. The geographic distribution of participants is added in the manuscript as per the question above. The frequency of concurrent conditions and the frequency of concomitant medication use is described in the supplement; however, it is acknowledged this provides limited information as to the general medical condition of the participants. No variable on general medical condition was derived from the reported conditions and medication-use for this study. The background comparator used in the study does not stratify their findings on maternal medical condition.
Indeed, no statistical comparison between the evaluable study population and the lost to follow-up population was done, but it is to be noted that the lost to follow-up population was small in this study (3.9%). |
6. The authors use the expression “not statistically significantly different” (Page 8, line 306) when comparing outcomes of the general US population. What statistical test was employed? |
Thank you, the language has been updated to say: The point estimate for the prevalence of MCMs in our study (1.9% [upper 95% CI 3.1%]) was similar to the background prevalence of 2.78%. |
7. Limitations of the study. The sample size was small to detect significant differences for many of the outcomes, and the absence of a control group is another important limitation as the authors reported. In addition, some outcomes are missing like the Apgar index or small for gestational age. Considering these circumstances, the generalization of the results may be problematic. |
That is agreed, and the limitations of this study are indeed described. The conclusion of the study is that there is no suggestion of a safety concern and the findings correspond to what is reported in other studies conducted in pregnant people. |
Reviewer 2 Report
I would like to recommend this manuscript for publication after minor revision:
1. As a big data clinical study, the authors should give where the data in the table are obtained, worldwide, or specific countries.
2. As an article, the data in the text is not enough. Please consider move part of the supporting materials to the main text, or change the article as a communication.
3. Are the obtained clinical data authorized by specific hospitals or patients?
4. Figure 1, why the legend concludes cited references? Is this figure cited from other references? If so, is the copyright permission obtained?
5. In the Author Contribution part, should J.v.B be J.V.B?
6. In the Introduction part, it is suggested to add some possible conjectures about the possible effects of vaccines on some patients with other underlying diseases or complications, such as intestinal diseases < Engineered Regeneration 2 (2022) 279-287.>.
Author Response
REVIEWER #2 |
AUTHOR RESPONSE |
1. As a big data clinical study, the authors should give where the data in the table are obtained, worldwide, or specific countries. |
This study used data reported by obstetric HCPs for individual participants. All data are collected in the United States. In line 165 the states in which the HCPs held their clinics are added. |
2. As an article, the data in the text is not enough. Please consider move part of the supporting materials to the main text, or change the article as a communication. |
The authors have reviewed the entirety of the text. After implementing the breadth of the reviewer suggested revision the paper is now comprehensive in scope. |
3. Are the obtained clinical data authorized by specific hospitals or patients? |
The full study was reviewed and approved by the Western Institutional Review Board (WIRB) and subjects consented to sharing their data with the study. |
4. Figure 1, why the legend concludes cited references? Is this figure cited from other references? If so, is the copyright permission obtained? |
The data in the figure is derived from the references cited, not the figure itself. There is no copyright issue. |
5. In the Author Contribution part, should J.v.B be J.V.B? |
The name is Josephine van Boxmeer (a Dutch name) and ‘van’ is written as a separate word in small caps. |
6. In the Introduction part, it is suggested to add some possible conjectures about the possible effects of vaccines on some patients with other underlying diseases or complications, such as intestinal diseases < Engineered Regeneration 2 (2022) 279-287.>. |
Although it is acknowledged that influenza vaccines are recommended for multiple different at-risk groups, this study focuses on pregnant people as a risk group for influenza infection. Nevertheless, reference to other at-risk groups has been added to the introduction section (lines 47 and 48). |
Round 2
Reviewer 1 Report
The authors have addressed all the indications of the reviewer.
Author Response
We are thankful to you for your peer review and suggestions on the manuscript.